# Horizontal CO_2_ Compensation in the Yangtze River Delta Based on CO_2_ Footprints and CO_2_ Emissions Efficiency

**DOI:** 10.3390/ijerph20021369

**Published:** 2023-01-12

**Authors:** Luwei Wang, Yizhen Zhang, Qing Zhao, Chuantang Ren, Yu Fu, Tao Wang

**Affiliations:** 1School of Geographical Sciences, Nanjing Normal University, Nanjing 210023, China; 2School of Tourism and Urban Management, Jiangxi University of Finance and Economics, Nanchang 330013, China

**Keywords:** CO_2_ footprints, CO_2_ emissions efficiency, CO_2_ trading, horizontal CO_2_ compensation

## Abstract

Purpose: In this study, we attempted to reduce the negative economic externalities related to Carbon Dioxide (CO_2_) emissions in the Yangtze River Delta region (YRD) and designed a cross-municipality responsibility-sharing mechanism. Methods: We estimated the municipal CO_2_ footprints in the YRD from 2000 to 2019 based on nighttime light data and measured CO_2_ emissions efficiency using a super slack-based measurement (super-SBM) model. Based on this, we designed a scenario of horizontal CO_2_ compensation among the YRD’s municipalities from the perspectives of both CO_2_ footprints and CO_2_ trading (CO_2_ unit prices in trading were determined based on CO_2_ emissions efficiency). Results: The results showed the following: (1) The CO_2_ footprints evolution of the YRD municipalities could be divided into four categories, among which, eleven municipalities showed a decreasing trend. Thirteen municipalities stabilized their CO_2_ footprints. Thirteen municipalities exhibited strong growth in their CO_2_ footprints, whereas four municipalities maintained a low level of slow growth. (2) Spatially, CO_2_ emissions efficiency evolved from a broad distribution of low values to a mosaic distribution of multi-type zones. (3) After 2011, the ratio of CO_2_ footprint compensation amounts to local Gross Domestic Product (GDP) in most municipalities was less than 0.01%, with its center of gravity shifting cyclically. It was appropriate to start charging the CO_2_ footprint compensation amounts after 2011, with a dynamic adjustment of 3 years. (4) After 2007, the supply–demand relationship of CO_2_ trading continued to deteriorate, and it eased in 2016. However, its operational mechanism was still very fragile and highly dependent on a few pioneering municipalities. Innovations: In this study, we designed a horizontal CO_2_ compensation mechanism from the binary perspective of CO_2_ footprints and CO_2_ trading. In this mechanism, the former determines the CO_2_ footprint compensation amounts paid by each municipality based on whether the CO_2_ footprint exceeds its CO_2_ allowance. The latter determines the CO_2_ trading compensation amounts paid by the purchasing municipalities based on their CO_2_ emissions efficiency. This system balances equity and efficiency and provides new ideas for horizontal CO_2_ compensation.

## 1. Introduction

The warming caused by excessive fossil energy consumption has led the international community to rethink the current economic growth model [1,2]. A ‘carbon neutrality’ approach to deep decarbonization is becoming a significant trend in the evolution of the global energy economy [3,4]. From the Kyoto Protocol to The Paris Agreement, the division of responsibility for the reduction of CO_2_ emissions has always been at the heart of the international approach to CO_2_ emissions [5]. Based on the constraints of the global CO_2_ emissions and the realistic requirements of domestic economic transformation, China is actively fulfilling its commitments under the National Autonomous Contribution framework. China has incorporated binding emissions-reduction targets into its long-term economic development plan and proposed the goal of ‘achieving carbon neutrality by 2060’ at the United Nations General Assembly in September 2020.

Inter-regional horizontal CO_2_ compensation has become a new governance model to promote synergistic regional economic development [6,7]. Although it is impossible to achieve a mandatory national CO_2_ emissions reduction, it is feasible for China to adopt the idea of regional pilot projects first, followed by national implementation. The YRD has the market base to establish a unified environmental rights and interests trading system. The cumulative spot trading volume of the Shanghai CO_2_ trading market reached 159 million tons in 2020, with the trading volume exceeding RMB 1.7 billion. The results achieved with CO_2_ trading have made it increasingly clear that CO_2_ emissions are property-rights-based and tradable. This has raised awareness and increased the capacity of neighboring municipalities to manage their CO_2_ assets. Secondly, Zhejiang and Jiangsu provinces have been working on emissions trading for over ten years and have a wealth of experience. In addition, Zhejiang and Anhui provinces have developed models of watershed protection to follow in terms of ecological compensation. Thus, it seems that the YRD is an essential test region for the horizontal CO_2_ compensation system in China.

The horizontal CO_2_ compensation system aims to compensate for the negative externality of CO_2_ emissions and to ensure that municipalities develop a low-carbon economy with fairness and efficiency. Some municipalities are developing economies at the cost of a welfare loss for the environments of neighboring municipalities. However, these municipalities do not have to bear this external cost, which results in the marginal cost to society being greater than the marginal cost to individuals, and which can be called a negative externality. The negative externality caused by CO_2_ emissions is deeply affecting climate change and is an urgent issue to be addressed regionally, nationally, and globally. For the YRD, the traditional competitive model of ‘competitive growth’ which emerged in the early days under the performance appraisal mechanism of the GDP promotion race became one of the institutional root causes of sloppy and distorted economic growth, which directly led to severe environmental pollution and energy consumption. With the transformation of the economic model and the improvement of the government’s governance capacity, the performance appraisal mechanism for Chinese municipal officials has gradually strengthened its assessments of energy consumption and CO_2_ emissions indicators. Horizontal CO_2_ compensation will become a long-term mechanism to promote the development of China’s low-carbon economy from top to bottom. It requires not only balancing the development of each municipality in the process of CO_2_ emissions reduction to avoid the further development of disparities among municipalities but also taking into account the differences in the CO_2_ emissions efficiency among municipalities to prevent the bullwhip effect. Therefore, in this study, we explored the issues of how to set municipal CO_2_ allowances based on CO_2_ footprints, how to set the CO_2_ unit prices in trading among municipalities according to their CO_2_ emissions efficiency, and how to design a horizontal CO_2_ compensation system among municipalities while considering the principles of equity and efficiency.

This paper enriches or expands on the existing studies in the following aspects. Firstly, due to the lack of energy consumption data at the municipal scale in China, most studies have explored CO_2_ footprints and CO_2_ emissions efficiency at the provincial scale, which could hinder the establishment of a joint responsibility mechanism for CO_2_ emission reduction among the municipalities of the YRD. Since the 1980s, Welch has been the first to verify the feasibility of using nighttime lighting data to estimate energy consumption, using eastern U.S. cities as the study target [8]. Subsequently, a series of studies confirmed the consistency between nighttime lighting values and energy consumption footprints, which provided the theoretical basis and methods for calculating the municipal CO_2_ footprints in the present study. However, relevant studies with municipalities in the YRD as a sample are still lacking, and this study has enriched this region of research. Secondly, we determined municipalities’ horizontal CO_2_ compensation system by considering their emission reduction potential and economic conditions. On the one hand, unlike the studies conducted by Hu et al. [9] and Yang et al. [10], we used the historical CO_2_ intensity method to determine municipal CO_2_ allowances. In this method, we calculated CO_2_ intensity based on municipal historical production data and CO_2_ emissions and reduced it year by year to determine the CO_2_ allowance for the following year. On the other hand, we considered the ratio of the CO_2_ footprint compensation amount to local GDP, as well as the stability of municipalities’ CO_2_ surplus and deficit statuses when determining the implementation time of the compensation system. Thirdly, most existing studies have determined the horizontal CO_2_ compensation amounts based on the municipal CO_2_ surplus and deficit status (CO_2_ footprints) [9,10]. However, the difference in municipal CO_2_ emissions efficiency makes the two sides of CO_2_ trading obtain significantly different economic values. Therefore, this study further expands the existing research by including the added value generated in CO_2_ trading into the horizontal CO_2_ compensation system.

## 2. Literature Review and Study Framework

The concept of a CO_2_ footprint originated from the ecological footprint concept proposed by William in 1992, which refers to the number of greenhouse gases (GHG) emitted through production and consumption activities. CO_2_ footprint calculation is one of the effective ways to evaluate GHG emissions, and there are currently three main calculation methods. The first is the life cycle assessment method (LCA), which mainly assesses the CO_2_ emissions of a product during its life cycle and is a bottom-up evaluation method based on process analysis [11,12]. The second is the input–output analysis method (I-O), which mainly uses the preparation of input–output tables (IOTs) for accounting, including the single-region (SRIO) and multi-region input–output method (MRIO) [13]. It is a top-down model based on input–output analysis [14]. The third is the Intergovernmental Panel on Climate Change (IPCC) coefficient method, which uses the national GHG inventories compiled by the IPCC and the corresponding emission coefficients to calculate CO_2_ emissions [15,16]. All three methods mentioned above have their applicability and limitations. Specifically, LCA considers both direct and indirect CO_2_ emissions during the system’s life cycle with high accuracy and is suitable for CO_2_ footprint calculation at the micro level for products. However, it involves significant obstacles in acquiring data at the macro level. In addition, it involves problems such as truncation errors in the delineation of system boundaries and subjectivity in the standardization and weighting processes [11,12]. I-O can comprehensively reflect the direct and indirect CO_2_ emission relationships of various sectors and can overcome the problem of the duplication or omission of calculations due to the complex production relationships among sectors. However, the currently compiled IOT exhibits a significant lag time, and the aggregation mode of sectors in the IOT may be different from those of the energy consumption data, leading to some errors in the calculation of CO_2_ footprints [14]. The IPCC coefficient method comprehensively examines the GHG emissions caused by the combustion of different energies. It is easy to obtain data using this method and it involves a relatively simple calculation process, which is suitable for CO_2_ footprint calculations in regard to energy consumption at all scales. However, it cannot cover implied indirect CO_2_ emissions [15]. Since it is hard to obtain IOTs for municipalities and to monitor all the substances and activities covered by each product life cycle within a municipality [17], scholars usually use the IPCC method to calculate municipal-scale CO_2_ footprints. It is worth noting that municipal-scale energy consumption data are also difficult to obtain. Fortunately, with the development of earth observation techniques, nighttime light data from DMSP/OLS and VIIRS/DNB provide essential aids for the process of downscaling, which his used to calculate municipal CO_2_ footprints. The relevant studies date back to the 1980s, when Welch was the first to validate the feasibility of using nighttime light data to estimate energy consumption, using eastern cities in the United States as the study target [8]. Subsequently, Elvidge et al. [18] and Doll et al. [19] further confirmed the correlation between nighttime light brightness values and CO_2_ footprints, providing the theoretical foundation and empirical examples for modeling CO_2_ footprints on the basis of nighttime light data. Recently, some scholars have also started to apply nighttime light data to analyze CO_2_ footprints, proposing a quantitative method to monitor municipal CO_2_ footprints from top to bottom [20,21,22].

Based on measurements of CO_2_ footprints, evaluating CO_2_ emissions efficiency has become a popular area of academic study. Scholars have successively proposed various measurement criteria, such as the CO_2_ index (CO_2_ emissions per unit of energy consumption), energy intensity (energy consumption per unit of GDP), CO_2_ intensity (CO_2_ emissions per unit of GDP), etc. It is worth noting that the above criteria are ‘single indicators’ in nature, ignoring the impact of indicator substitution effects on CO_2_ reduction activities. As data envelopment analysis (DEA) methods become more established in the environmental field, more studies have shown that energy has to be integrated with production indicators such as capital and labor in order to obtain accurate outputs [23,24,25]. It is more reasonable to explore the issue of CO_2_ emissions efficiency in a full-indicator framework [26]. Traditional DEA models focus only on the desired outputs of economic activity and ignore the undesired ones, which biases the results. As a result, scholars have improved these models [27,28]. For example, Teng et al. introduced CO_2_ sinks as an exogenous variable into a modified dynamic SBM model based on China’s afforestation area data to assess provincial CO_2_ emissions efficiency [29]. Zhao et al. used the epsilon-based measure (EBM)-DEA model to estimate the CO_2_ emissions efficiency of the transport sector in Chinese provinces. They further explored the impact of transport structure, technological progress, and urbanization level on CO_2_ emissions efficiency [30]. Wu et al. used an extended SBM model to analyze the spatio-temporal evolution of provincial CO_2_ emissions efficiency. They used *σ* and spatial *β*-convergence tests to explore whether there was a catch-up trend in CO_2_ emissions efficiency among provinces [31]. Xie et al. used a super-SBM model to assess the CO_2_ emissions efficiency of 59 countries. They also used national panel quantile regressions to explore the multiple effects of technological progress on CO_2_ emissions efficiency [32].

Horizontal CO_2_ compensation can help to coordinate regional environmental protection measures and achieve ‘carbon neutrality’ [33]. CO_2_ compensation studies can be traced back to the 1950s, when the U.S., France, Mexico, and Costa Rica explored ‘ecological compensation’ as a method of environmental management. ‘Ecological compensation’ originally referred to the payments for ecosystem/environmental services (PES) made by environmental services consumers to suppliers for access to natural resource services [6,34,35]. Recently, PES has been applied extensively to transboundary pollution management, and CO_2_ compensation has become an emerging area of inquiry [9]. Scholars have focused on CO_2_ compensation studies from three main perspectives: theory, methods of measuring compensation amounts, and applications. (i) The public goods theory, ecological capital theory, and ecological justice theory together form the philosophical foundation and theoretical cornerstone of CO_2_ compensation. Public goods theory mainly emphasizes the negative externalities of CO_2_ emissions, meaning that one economic agent does not fully bear the costs of its emissions, causing welfare losses for other economic agents [36,37]. Scholars have attempted to construct a system that internalizes all externalities, leading to a debate on the Pigouvian CO_2_ tax and the Coasean approach. Ecological capital theory emphasizes the capital properties of CO_2_ emissions rights. More and more countries, organizations, and enterprises are realizing the value embedded in CO_2_ emissions rights. A CO_2_ trading market based on allowances has gradually developed, giving rise to CO_2_ emissions rights futures, options trading, and other financial products [38]. The ecological justice theory emphasizes the need for CO_2_ emissions behavior to be organically integrated with social justice, working to unlock the international climate dilemma of collective action [39,40]. (ii) The main methods for measuring CO_2_ compensation amounts include the ecosystem service value method [34], the cost analysis method [41], the willingness assessment method [42], and the footprints method [9]. For example, Cui et al. used the ecosystem service value method to investigate the impact of human activities on CO_2_ emissions from wetlands. They calculated the value of CO_2_ management in the Zoige wetland using a local trading price for CO_2_ reductions. Their method was consistent with the economic theory that value determines the price, and their accounting results were relatively objective [43]. Engel et al. used a cost analysis to model decisions on representative land. They designed payment schemes linked to the price of an agricultural crop (soybean) and the price of CO_2_ emissions by comparing the costs and future returns of interchanging the two land types of forest and agriculture. Their method considered the direct input costs for ecological conservation and the development opportunity costs [41]. Albertini et al. applied the willingness assessment method to assess the payment criteria for Italian and Czech households by conducting web-based interviews to determine participants’ willingness to pay for CO_2_ emissions from energy consumption. This method considered CO_2_ emitters’ willingness and facilitated the implementation and diffusion of CO_2_ compensation systems [44]. Hu et al. used the CO_2_ footprint and carrying capacity calculation methods to estimate the CO_2_ source and sink functions and spatial distribution of eleven municipalities in Jiangxi Province. They determined the horizontal ecological compensation amounts based on municipalities’ CO_2_ surplus and deficit statuses [9]. The footprints method is more applicable to calculating the horizontal CO_2_ compensation amounts than the other three methods. The main reason for this is that the statistical standards for energy consumption data in different regions are relatively consistent. The cost of ecological compensation can be conveniently and uniformly accounted for across municipalities based on CO_2_ footprints. (iii) Application studies of CO_2_ compensation are usually carried out for specific projects, such as CO_2_ compensation for engineering construction [45], reservoir development [46], forestry and grass types [47], etc. In contrast, horizontal CO_2_ compensation is still in the exploration stage, and studies on inter-provincial and inter-municipal CO_2_ compensation standards and models require further enrichment [48].

The remainder of this paper is structured as follows. Section 3 presents the data sources and methods. The methods include a method estimating CO_2_ footprints based on nighttime light data, a super-SBM model for calculating the CO_2_ emissions efficiency, and a method of calculating horizontal CO_2_ compensation amounts. Section 4 first presents trends in municipal CO_2_ footprints and their spatial correlation. Secondly, it describes the spatio-temporal differences in CO_2_ emissions efficiency. Finally, scenarios of horizontal CO_2_ compensation among municipalities are discussed from the two perspectives of CO_2_ footprints and CO_2_ trading (Figure 1). Section 5 summarizes and discusses the conclusions and recommendations.

## 3. Data and Methods

### 3.1. Data Collection

(1) DMSP/OLS stable annual light data for 2000–2013 were obtained from the National Oceanic and Atmospheric Administration-National Geophysical Data Center (NOAA/NGDC, http://www.ngdc.noaa.gov/eog/dmsp.html, accessed on 13 May 2022). Monthly VIIRS/DNB light data for 2012–2019 were obtained from the Earth Observation Group (EOG, https://eogdata.mines.edu/products/vnl, accessed on 13 May 2022). MOD13A2-EVI vegetation indices were obtained from the National Aeronautics and Space Administration. Impervious surface data were derived from the Global Urban Boundary Dataset. (i) In this study, we applied continuum correction to DMSP/OLS data [49] and desaturated them using enhanced vegetation indices [50]. (ii) We converted the VIIRS/DNB data into annual sequence data, extracted the image elements with *DN* ≠ 0 in DMSP-OLS to generate a mask, and performed noise removal on the VIIRS/DNB data. (iii) After the processing, individual image element outliers were substituted with neighboring image element maxima. (iv) The high-precision extraction of information on built-up areas is a prerequisite for conducting CO_2_ footprint simulations. The optimal segmentation threshold was set to *DN* = 2.9 (via the mutation detection method) to extract the built-up area information [51]. The extraction accuracy (95.50%) was calculated based on the built-up area extracted from the impervious surface, and the results met the requirements. (v) The relationship between the 2013 DMSP/OLS and VIIRS/DNB data was quantified using an S-shaped function so that both had an identical spatial resolution and similar radiometric characteristics [52]. The derived relationship enabled us to generate globally coherent nighttime light raster data for 2000–2019, which masked the extraction of nighttime light data from Chinese municipalities.

(2) Energy data were obtained from the *China Energy Statistical Yearbook* and from provincial and municipal statistical yearbooks, covering eight types of energy: coal, coke, crude oil, gasoline, paraffin, diesel, energy oil, and natural gas. To measure the average energy, we calculate low-level calorific values according to the *General Rules for Calculating Comprehensive Energy Consumption* (GB/T2589-2008). The carbon content per unit calorific value and the carbon oxidation rate are referred to in the *Guidelines for the Preparation of Provincial Greenhouse Gas Inventories* (National Development and Reform Commission Climate (2011), No. 1041).

(3) CO_2_ emissions efficiency input indicators and desired output indicators are from the *China Urban Statistical Yearbook*, *Shanghai Statistical Yearbook*, *Jiangsu Statistical Yearbook*, *Zhejiang Statistical Yearbook*, and *Anhui Statistical Yearbook*.

### 3.2. Methods

#### 3.2.1. Estimation of the CO_2_ Footprints

(1) Estimation of provincial CO_2_ footprints: we calculated CO_2_ footprint coefficients (*A_m_*) for coal (1.900 kg-CO_2_/kg), coke (2.860 kg-CO_2_/kg), crude oil (3.020 kg-CO_2_/kg), gasoline (2.925 kg-CO_2_/kg), paraffin (3.072 kg-CO_2_/kg), diesel (3.096 kg-CO_2_/kg), energy oil (3.171 kg-CO_2_/kg), and natural gas (2.162 kg-CO_2_/kg) based on the average energy low-level calorific value, the carbon content per unit calorific value, and the carbon oxidation rate. The provincial CO_2_ footprints (*C_p_*) were calculated according to (Equation (1)).
(1)$$C_{p}=\sum_{m=1}^{M=8}E_{pm}\times A_{m}$$
where *E_pm_* is the consumption of the *m*th energy in the *p*th provincial district, *m* = 1, 2, 3 … *M*.

(2) Fitting of municipal CO_2_ footprints: The fitted curve (Equation (2)) was determined based on the total brightness values of provincial nighttime light (*TNLI_p_*) and the provincial CO_2_ footprints (*C_p_*). Then, the municipal CO_2_ footprints (*C_i_*) (Equation (3)) were estimated based on the regression parameter (*ω*), the power law parameter (*τ*) in Equation (2), and the total brightness value of municipal nighttime light (*TNLI_i_*).
(2)$$C_{p}=\omega \times {TNLI}_{p}^{\tau }$$
(3)$$C_{i}=\omega \times {TNLI}_{i}^{\tau }$$

(3) Accuracy checking of municipal CO_2_ footprints. We checked the model accuracy in two ways. The first method was to check the fitting of Equation (2). *R*^2^ was a critical indicator for comparing the fitting of the model. The higher the *R*^2^, the more reliable the regression results. The second method was to use municipalities with energy statistics as a test sample. We computed the actual CO_2_ footprints based on the energy statistics provided for the test sample. At the same time, the total brightness value of the nighttime light for the test sample was input into Equation (3) with the fitted CO_2_ footprints as the output for the same period. The deviation between the actual and fitted CO_2_ footprints was compared to check the accuracy of the results.

#### 3.2.2. Estimation of CO_2_ Emissions Efficiency

(1) The super-SBM model is a popular method to determine CO_2_ emissions efficiency. Its mathematical logic is to compare the input–output systems of all municipalities and screen out the Pareto optimal solution with the lowest input and undesired output and the highest desired output to constitute the frontier surface. Slack variables are also incorporated into the objective function to calculate the deviation distance from each municipality to the production frontier surface and use it to evaluate the municipal CO_2_ emissions efficiency. The most significant advantage of the super-SBM model is that it solves the input–output slackness problem and the efficiency decomposition problem in the presence of undesired outputs without the need to estimate the parameters in advance or to make weighting assumptions. It avoids the influence of human subjective factors and broadens the applicable constraints of the model. In addition, it can overcome the previous drawback of not being able to compare the efficiency values of multiple municipalities when they are 1 [53,54]. Based on these characteristics, we applied the super-SBM model to measure the municipal CO_2_ emissions efficiency (*ρ_it_*) (Equation (4)) [55].

(2) We constructed the input–output indicator system based on the principle that the number of input and output indicators should not exceed 1/3 of the number of decision-making units. (i) Rapid economic development is highly dependent on the scale of investment growth, especially in regard to enormous government-led investments in infrastructure. Strong geographical inertia is essential in the choice of location for large, energy-intensive, installation-based industries. Therefore, we calculated the social fixed-asset investment volume as capital inputs using the perpetual inventory method, with 2000 as the base year [56]. (ii) The material production process is the process of labor acting on productive resources. A large employment population attracts a concentration of labor-intensive industries, which can bring about problems such as a decrease in environmental air quality, while promoting economic development. Therefore, we chose year-end employment as a labor input [56,57]. (iii) The crucial aspect in developing a low-carbon economy is saving energy without curtailing economic output. Energy utilization efficiency has a material impact on the decarbonization of municipal economies. Therefore, based on the availability of data and the avoidance of covariance with undesired outputs, we used the total social electricity consumption as an energy input [56,58]. (iv) Science and technology expenditure drives CO_2_ capture and storage technologies from the experimental stage to the diffusion stage. It is a fundamental force in the transformation of economic models. Therefore, we used science and technology expenditure as a measure of municipalities’ technology inputs. (v) GDP (converted to constant 2000 prices) reflects the overall level of municipal economic development [56,57,58]. Retail sales of social consumer goods reflect the material living standards of citizens and the purchasing power of social goods. Both are positive indicators of the feasibility of the design path of decarbonization. Therefore, in this study we regarded them as desired outputs of CO_2_ reduction activities. (vi) Fossil energy combustion is the primary source of carbon increase in the atmosphere, and CO_2_ footprints are also the core of this paper. Therefore, CO_2_ footprints were considered undesired outputs of CO_2_ reduction activities [56,57,58].
(4)$$\rho_{i}^{t}=\min\frac{1-\frac{1}{A}\sum_{a=1}^{A}\frac{s_{ia}^{t\text{-}in}}{x_{ia}^{t}}}{1+\frac{1}{B+C}(\sum_{b=1}^{B}\frac{s_{ib}^{t\text{-}out}}{y_{ib}^{t}}+\sum_{c}^{C}\frac{s_{ic}^{t\text{-}out^{\prime }}}{z_{ic}^{t}})}$$

The constraints were as follows.
$$x_{ia}^{t}\geq \sum_{i=1,i\neq j}^{I}\left(x_{ia}^{t}\lambda_{i}-s_{ia}^{t\text{-}in}\right),\;1-\frac{1}{B+C}(\sum_{b=1}^{B}\frac{s_{ib}^{t\text{-}out}}{y_{ib}^{t}}+\sum_{c=1}^{C}\frac{s_{ic}^{t\text{-}out^{\prime }}}{z_{ic}^{t}})>0$$
where *x_ia_^t^*, *y_ib_^t^,* and *z_ic_^t^* are the *a*th input indicators, the *b*th desired output indicator, and the *c^th^* undesired output indicator for the *i*th municipality in year *t*, respectively. *s_ia_^t-in^*, *s_ib_^t-out^,* and *s_ic_^t-out^* are the slack values of the above three indicators, respectively.

#### 3.2.3. Horizontal CO_2_ Compensation Model

Horizontal CO_2_ compensation is an institutional arrangement that relies on economic instruments to regulate the interests of relevant municipalities to solve the negative externalities of CO_2_ emissions. We conducted a study in terms of CO_2_ footprints and CO_2_ trading (with CO_2_ unit prices in the trading process determined by CO_2_ emissions efficiency). The former concept was focused on energy consumption behavior, and this determined the CO_2_ compensation amounts paid by each municipality based on whether their CO_2_ footprints exceed their CO_2_ allowances. The compensation amounts were closely related to the municipal CO_2_ footprints. The latter concept focused on CO_2_ trading activities, and this involved charging purchasing municipalities a fee based on the economic benefits generated in transferring CO_2_ emissions rights among municipalities. In this way, we designed a horizontal CO_2_ compensation model from a binary perspective. In terms of equity (whether CO_2_ footprints exceeded CO_2_ allowances) and efficiency (whether the CO_2_ emissions efficiency reached the average level), we classified municipalities into four types: ‘high-efficiency and low-emissions’, ‘high-efficiency and high-emissions’, ‘low-efficiency and high-emissions’, and ‘low-efficiency and low-emissions’ municipalities (Figure 1).

From the perspective of CO_2_ footprints: The YRD exhibits uneven economic development. ‘One-size-fits-all’ instruments will lead to inefficient and unfair horizontal CO_2_ compensation. Therefore, in this study, we followed the principle of a universal levy and additional adjustments.

Firstly, we used the historical CO_2_ intensity method [59] to allocate CO_2_ allowances to municipalities (Equation (5)).
(5)$$q_{it}=\vartheta \cdot {GDP}_{it}$$
where *GDP_it_* is the GDP of the *i*th municipality in year *t*. *ϑ* is a base value for the historical CO_2_ intensity. When the CO_2_ intensity continues to rise or fall in years *t* − 3 to *t* − 1 and the cumulative change exceeds 30%, *ϑ* is taken as the CO_2_ intensity in year *t* − 1. Otherwise, *ϑ* takes the average value of the CO_2_ intensity in years *t* − 3 to *t* − 1.

Then, a differentiated CO_2_ footprint income and expenditure coefficient (CFIE) was developed, including a baseline CFIE (*δ_q_*) and an additional CFIE (*δ_∆q_*). For CO_2_-deficit municipalities, CO_2_ footprints were paid at *δ_q_* for the portion not exceeding the CO_2_ allowances, and the excess was paid at *δ*_∆*q*_. For CO_2_-surplus municipalities, CO_2_ footprints were paid at *δ_q_*, whereas these municipalities are compensated at *δ*_∆*q*_ according to their remaining CO_2_ space. The baseline CO_2_ footprint compensation amounts paid at *δ_q_* are an essential component of the financial transfer among municipalities (top tier) under the YRD emissions reduction responsibility-sharing mechanism. These ultimately come from the CO_2_ tax levied by the municipal government on local enterprises or residents (the bottom tier). Therefore, *δ_q_* should be relatively consistent with the observed value of the CO_2_ tax rate (RMB 40/ton). In addition, we set the value of *δ_∆q_* based on the growth curve function (Equation (6)) [60].
(6)$$\delta_{\Delta q}=(1+\frac{{GDP}_{it}\times \rho_{i}^{t\varphi }}{\left(1+e^{-f}\right)\sum_{i=1}^{I}\left({GDP}_{it}\times \rho_{i}^{t\varphi }\right)})\delta_{q}$$
where *f* is the average value of the Engel coefficient of the 41 municipalities in year *t*. *φ* is taken to be a constant in order to avoid the excessive influence of CO_2_ emissions efficiency on *σ_t_*.

Finally, we calculated the CO_2_ footprint compensation amounts *A_i′t_* for the *i*′th CO_2_-deficit municipalities (Equation (7)) and *A_i_*_″*t*_ for the *i*″th CO_2_-surplus municipalities (Equation (8)), respectively.
(7)$$A_{i^{\prime }t}=\delta_{q}\times C_{i^{\prime }t}+\delta_{\Delta q}(C_{i^{\prime }t}-q_{i^{\prime }t})$$
(8)$$A_{i^{\prime \prime }t}=\delta_{q}\times C_{i^{\prime \prime }t}-\delta_{\Delta q}(q_{i^{\prime \prime }t}-C_{i^{\prime \prime }t})$$

(2) From the perspective of CO_2_ trading with a price mechanism determined by CO_2_ emissions efficiency: As a valuable asset and strategic resource with geographical attributes, CO_2_ emissions rights are a unique intangible commodity. The inequality in CO_2_ emissions efficiency leads to differences in the economic benefits obtained per unit of CO_2_ emissions among municipalities. On the one hand, CO_2_ trading follows a self-regulatory mechanism for pricing. The higher a municipality’s CO_2_ emissions efficiency, the higher the unit prices of the CO_2_ emissions rights (CO_2_ unit prices) it sells, and vice versa. On the other hand, CO_2_ emissions rights may generate added value during transfers among municipalities, from which the purchasing municipality benefits. In this study, we calculated the input amounts (*A*′*_b_*) based on the CO_2_ unit prices in the selling municipality and the output amounts (*A*′*_s_*) based on the CO_2_ unit prices in the purchasing one. Equation (9) was used to calculate the CO_2_ trading compensation amounts (*A*′_*i*′*t*_) to be paid by the purchasing municipality.
(9)$${A^{\prime }}_{i^{\prime }t}={A^{\prime }}_{s}-{A^{\prime }}_{b}=(\rho_{i^{\prime }}^{t}\cdot pri)\cdot \Delta c\cdot \delta^{\prime }-(\rho_{i^{\prime \prime }}^{t}\cdot pri)\cdot \Delta c\cdot \delta^{\prime }$$
where *pri* is the CO_2_ unit price, which was set to RMB 36.36/ton [9]. (*ρ*_*i*′_*^t^·pri*) and (*ρ_i_*_″_*^t^·pri*) are the CO_2_ unit prices in the selling and purchasing municipalities, respectively. ∆*c* is the CO_2_ trading volume. *δ′* is the value-added coefficient for CO_2_ turnover based on the VAT rate for intangible asset transfers (6%).

To maximize the economic benefits for the entire YRD, the government plays the following regulatory role (Figure 1). (i) The government strictly controls the sales of ‘high-efficiency and low-emissions’ municipalities at a suppressed price in CO_2_ trading. (ii) The government uses part of the CO_2_ compensation amounts in the form of energy savings and environmental protection investments to the ‘low-efficiency and high-emissions’ and ‘low-efficiency and low-emissions’ municipalities to help these poorly performing municipalities in an intelligent way. Part of these amounts are in the form of financial subsidies to guide ‘high-efficiency and low-emissions’ and ‘low-efficiency and low-emissions’ municipalities to prioritize the CO_2_ emissions needs of ‘high-efficiency and high-emissions’ municipalities. ‘High-efficiency and high-emissions’ municipalities compete for priority purchase rights through competition for CO_2_ emissions efficiency among themselves. (iii) The government uses penalties to discourage municipalities with high CO_2_ emissions efficiency from selling CO_2_ emissions rights to municipalities with low CO_2_ emissions efficiency (∆*c* < ∆*c_G_*) (Figure 2). Accordingly, in this study we have designed a CO_2_ trading market.

When $\sum_{i^{\prime }}^{I^{\prime }}\left(P_{i^{\prime }t}-c_{i^{\prime }t}\right)<\sum_{i^{\prime \prime }}^{I^{\prime \prime }}\left(c_{i^{\prime \prime }t}-P_{i^{\prime \prime }t}\right)$, competition mainly occurs among selling municipalities in a CO_2_ trading market with a surplus of total CO_2_ allowances, and the overall economic benefits brought about by CO_2_ trading are $E\cot=\int_{\Delta c=0}^{\Delta c=\sum_{i^{\prime }}^{I^{\prime }}\left(P_{i^{\prime }t}-c_{i^{\prime }t}\right)}f^{\prime }(\rho_{i^{\prime }}^{t})-f^{\prime \prime }(\rho_{i^{\prime \prime }}^{t})d\Delta c$. When $\sum_{i^{\prime }}^{I^{\prime }}\left(P_{i^{\prime }t}-c_{i^{\prime }t}\right)>\sum_{i^{\prime \prime }}^{I^{\prime \prime }}\left(c_{i^{\prime \prime }t}-P_{i^{\prime \prime }t}\right)$, competition mainly occurs among purchasing municipalities in a CO_2_ trading market with a shortage of total CO_2_ allowances, and the overall economic benefits brought about by CO_2_ trading are $E\cot=\int_{\Delta c=0}^{\Delta c=\sum_{i^{\prime \prime }}^{I^{\prime \prime }}\left(c_{i^{\prime \prime }t}-P_{i^{\prime \prime }t}\right)}f^{\prime }(\rho_{i^{\prime }}^{t})-f^{\prime \prime }(\rho_{i^{\prime \prime }}^{t})d\Delta c$ (Figure 2).

## 4. Results

### 4.1. Spatio-Temporal Evolution of CO_2_ Footprints

#### 4.1.1. Accuracy Checking of Municipal CO_2_ Footprints

After comparing the accuracy obtained in several experiments, we selected 30 provincial districts (excluding Tibet, Hong Kong, Macao, and Taiwan) to fit the total brightness values of nighttime light and CO_2_ footprints. The fitted curves were determined based on three sets of panel data during the periods 2000–2006, 2007–2013, and 2014–2019 to estimate the CO_2_ footprints of 275 municipalities. We extracted the CO_2_ footprints of 41 municipalities in the YRD. Table 1 and Figure 3 show that all had *R*^2^ values higher than 0.7, and all relative errors were 5.013–5.209%, which means that the estimation results had good accuracy.

#### 4.1.2. Evolutionary Characteristics of the CO_2_ Footprints

Overall, the growth in energy demand in the YRD has led to a rise in CO_2_ footprints, from 745,919 million tons in 2000 to 1,881,457 million tons in 2019. In particular, from 2004 to 2011, the growth rate was the fastest, with an average annual growth rate of 8.69%. The pressure to reduce emissions is increasing. The growth rate in CO_2_ footprints from 2012 to 2019 has converged, with the average annual growth rate falling to 0.49%. The effectiveness of CO_2_ reduction continues to emerge.

Figure 4 shows the trends observed in the municipal CO_2_ footprints. Eleven municipalities, such as Shanghai (12,000–13,500 million tons), Suzhou (J.S.) (12,000–15,000 million tons), and Ningbo (7500–10,000 million tons), were in the CO_2_ peak range between 2010 and 2014, after which, they showed a decline. This indicates that the energy consumption structure of these municipalities has shown clean and low-carbon characteristics, and the effectiveness of CO_2_ reduction is beginning to show. Since 2014, the CO_2_ footprints of thirteen municipalities, including Hangzhou, Shaoxing, and Hefei, have stabilized. This indicates that the energy consumption of these municipalities has entered a period of ‘steady-to-moderate’ adjustment and that their economic development and industrial structures are undergoing a process of quantitative to qualitative change. The CO_2_ footprints of thirteen municipalities, including Fuyang and Bozhou, exhibited a vigorous increase. This is partly due to energy savings from technological progress being compensated for by additional energy consumption from economic growth. On the other hand, it is due to CO_2_ leakage from inter-municipality product trade and industrial transfers. Huangshan, Chizhou, Zhoushan, and Tongling, the CO_2_ footprints of which grew slowly and only crossed the 10 million tons threshold after 2010, have long maintained a robust supply of ecological products in the YRD.

Spatially, CO_2_ footprints decreased from the core of Shanghai to the marginal regions, with a significant spatial gradient. Figure 5 shows an inverted U-shaped development in the Moran’ I based on global spatial autocorrelation analysis. Between 2000 and 2008, CO_2_ emissions showed a high–high (HH) and low–low (LL) positive spatial agglomeration pattern, with the Moran’s I showing an upward trend from 0.338 to 0.490. This indicates that municipalities tended to focus on energy development and transport construction to improve the investment environment. Energy storage conditions and supply prices became an essential consideration standard for many enterprises to select locations during that period, leading to the spatial concentration of energy-consuming economic activities. After 2009, the proportion of high–low (HL) and low–high (LH) heterogeneous agglomeration patterns increased, and the Moran’s I showed a downward trend to 0.301. This indicates that people’s awareness of environmental protection has increased, and the type and amount of energy utilized affect the image of enterprises and the public’s acceptance of industrial activities. Many energy-intensive enterprises have either upgraded or moved out of their original locations. Figure 6 shows that CO_2_ footprints have club convergence characteristics based on local spatial autocorrelation analysis. The spatial distribution of HH agglomeration patterns (Shanghai, Suzhou (J.S.), Nantong, etc.) and LL agglomeration patterns (Chizhou, Huangshan, Tongling, etc.) was stable. This indicates that the spatial stability of the CO_2_ footprint agglomeration pattern in the YRD formed a local spatial pattern of ‘the lower is always lower, and the higher is always higher’. This is mainly because of the neighborhood and spatial spillover effect of CO_2_ emissions. The policy measures and strategic actions adopted by surrounding municipalities in terms of economic development and industrial layout have effects related to imitating, learning, and competing. There is convergence or complementarity in economic structures among neighboring municipalities.

### 4.2. Spatio-Temporal Evolution of CO_2_ Emissions Efficiency

Overall, the average value of CO_2_ emissions efficiency in the YRD has shown a fluctuating upward trend, with a significant fluctuation between 2008 and 2013. The main reason for this is that in 2008, China introduced a series of measures to expand domestic social consumption demand, such as the ‘package’, in response to the international financial crisis, which led to a large amount of capital flowing to energy-intensive industries. This led to a decline in CO_2_ emissions efficiency for two consecutive years in 2009 and 2010. In 2011, the outline of the 12th Five-Year Plan included CO_2_ intensity as a binding target. The YRD has responded positively to constructing a low-carbon economy, and CO_2_ emissions efficiency has been significantly improved, with the average value rising from 0.642 in 2010 to 0.739 in 2013.

Specifically, in this study we classified CO_2_ emissions efficiency into five levels (Figure 7): extreme low-value (0.201–0.400), low-value (0.401–0.700), medium-value (0.701–1.000), high-value (1.001~1.300), and extreme high-value (1.301–1.700). Between 2000 and 2006, CO_2_ emissions efficiency was generally low. Except for Nantong (*ρ*_2003_ = 1.079 and *ρ*_2005_ = 1.021), Jinhua (*ρ*_2006_ = 1.048), and Jiaxing (*ρ*_2006_ = 1.068), which reached an effective frontier in individual years, all other municipalities exhibited a non-effective status (*ρ_t_* < 1). The low-value areas were the most widely distributed. The medium-value areas evolved from a discrete point shape in 2000 to an inverse S-shape in 2006. Southern Jiangsu and the Hangzhou Bay region became growth hotspots, with Wuxi and Shaoxing showing slower growth rates and lagging behind surrounding municipalities in CO_2_ emissions efficiency. Between 2007 and 2013, the CO_2_ emissions efficiency of more municipalities reached the efficiency frontier. The low-value areas contracted further, and the medium-value areas expanded to become the most widely distributed type. The overall spatial pattern of CO_2_ emissions efficiency was high in the east and low in the west, with a symmetrical north–south distribution. Between 2014 and 2019, the extreme high-value and high-value areas were distributed along the Shanghai–Wuhan and Zhuji–Yongjia highway axes, with the formation of a backbone structure of Shanghai–Nanjing and Shanghai–Hangzhou–Ningbo. At the same time, some of the medium-value areas degenerated into low-value areas. Spatial homogenization evolved into spatial fragmentation with a mosaic driven by multiple cores.

### 4.3. Horizontal CO_2_ Compensation Scenarios

#### 4.3.1. Horizontal CO_2_ Compensation Based on CO_2_ Footprints

Each municipality needs to take responsibility for CO_2_ reduction and share the cost of environmental protection through financial payments to compensate its CO_2_ footprints. In this study, we used the historical CO_2_ intensity method to allocate CO_2_ allowances to municipalities. The CO_2_ emissions rights within these allowances maintain the sustainable development of each municipality’s economy, so each municipality pays its baseline CO_2_ footprint compensation amounts at a lower baseline CFIE (*δ_q_*). Figure 8a shows that the curve expressing the baseline CO_2_ footprint compensation amounts evolved from a general-normality to a standard-normality distribution. Namely, this curve flattened out from N~(884.228, 669.206^2^) in 2003 to N~(1804.084, 853.361^2^) in 2019. This suggests that the baseline compensation amounts increased significantly between 2003 and 2019, whereas the gap in the baseline compensation amounts among municipalities continued to narrow. This indicates that the baseline compensation amounts increases the absolute cost of emission reduction for municipalities, which is conducive to forcing municipalities to form a self-restraint mechanism, but it is difficult to have an effect on the relative cost of emission reduction. Therefore, it is necessary to pay the additional compensation amounts in case of exceeding the CO_2_ allowance.

To curb the negative economic externalities of CO_2_ emissions, CO_2_-deficit municipalities need to pay additional amounts for their CO_2_ footprints at a higher additional CFIE (*δ*_∆*q*_). At the same time, these amounts are used to compensate CO_2_-surplus municipalities at the same rate. Figure 8a shows that the curve expressing additional CO_2_ footprint compensation amounts evolved from a single peak to double peaks, with the peak gradually moving away from ‘y = 0’, indicating the evolution of the degree of CO_2_ surplus and deficit in the YRD towards polarization. In 2003, most municipalities did not exceed their CO_2_ allowances, and the additional CO_2_ footprint compensation amounts were concentrated around ‘y = 0’. Since then, some municipalities have shown increasing CO_2_ deficits, whereas others have gradually left more space for CO_2_ emissions. Between 2010 and 2019, the additional CO_2_ footprint compensation amounts paid by CO_2_-deficit municipalities and the amounts returned to CO_2_-surplus municipalities evolved towards a normal distribution, with symmetrical changes between the two. The difference in the additional CO_2_ footprint compensation amounts paid among CO_2_-deficit municipalities is small, and its curve is short and fat. However, the difference in the amounts returned among CO_2_-surplus municipalities is larger and its curve is thin and high.

Securing economic growth is a prerequisite for a low-carbon transition. Figure 8b shows that the ratio of the CO_2_ footprint compensation amounts (baseline compensation amounts + additional compensation amounts) to GDP in most municipalities from 2003 to 2011 was more than 0.01%. This suggests that the CO_2_ footprint compensation amounts would impose a disproportionate burden on local enterprises. Since 2011, the ratio of the CO_2_ footprint compensation to GDP has been decreasing. Except for some municipalities in Anhui, all others are capable of synergizing CO_2_ emissions reductions with economic development. Therefore, it is appropriate for the YRD to introduce CO_2_ footprint compensation amounts after 2011. The process requires compensation for some municipalities to reduce the distortionary effect of the CO_2_ footprint compensation amounts.

A horizontal CO_2_ compensation system based on CO_2_ footprints requires both stability in the long term and volatility in the short term. Figure 9 shows that before 2011, some municipalities in the Huaihai Urban Cluster and the Hefei Metropolitan Area frequently shifted from surplus to deficit status, with CO_2_ compensation zones increasingly shrinking. If a horizontal CO_2_ footprint compensation system were to be imposed in this period, it would increase the cost of policy implementation and governmental regulation. After 2011, CO_2_-surplus municipalities were mainly located in the marginal regions, and the status of CO_2_ surpluses and deficits remained stable over time. Additionally, the migration trajectory of the center of gravity of CO_2_ footprint compensation amounts showed that it oscillated back and forth in a southeast–northwest direction with irregularities between 2003 and 2010. After 2011, it shifted in a northwest direction, with the distance moved increasing every three years. The standard deviation ellipses in the 2011–2013, 2014–2016, and 2017–2019 periods were largely overlapping. Therefore, a horizontal CO_2_ footprint compensation system with a three-year adjustment period after 2011 could not only lead to savings in the cost of policy implementation but also help to avoid the problem of ‘incorrect collection and omission’ caused by delays in policy implementation and untimely feedback.

#### 4.3.2. Horizontal CO_2_ Compensation Based on CO_2_ Trading with a Price Mechanism Determined by CO_2_ Emissions Efficiency

Figure 10a shows that between 2003 and 2007, the CO_2_ surplus/deficit differential fluctuated around zero. This indicates that some municipalities had remaining CO_2_ space to meet the excess emissions of CO_2_-deficit municipalities and that the supply and demand in the CO_2_ market were relatively balanced. After 2007, the CO_2_ market was in short supply for a long time, reaching the lowest level in 2011 and 2016, with a significant imbalance between supply and demand. After 2016, the contradiction between supply and demand in the CO_2_ market gradually eased. The gap between supply and demand rebounded from −41,685,600 tons in 2016 to −12,090,600 tons in 2019.

The CO_2_ emissions efficiency of both sides of the transaction shows that most of the purchasing municipalities had significantly higher CO_2_ emissions efficiency than the selling ones. The purchasing municipalities tended to have high GDP and low CO_2_ emissions per unit of input and output indicators, whereas the selling ones tended to have the opposite. Therefore, the CO_2_ emissions space left by the selling municipalities for the purchasing ones increased the economic added value of CO_2_ emissions. In particular, the positive economic benefits increased rapidly after 2012, from RMB 105,619.81 million to RMB 383,468.02 million in 2019 (Figure 10b). The operation of a ‘you reduce emissions and I pay’ CO_2_ trading strategy would not only optimize the allocation of CO_2_ emissions rights but also improve the overall economic benefits of low-carbon development in the YRD.

Under government regulation, municipalities with higher CO_2_ emissions efficiency would achieve higher economic benefits per unit of CO_2_ emissions by purchasing the emissions rights of other municipalities. Following the principle of ‘I gain and I pay’, the purchasing municipalities pay CO_2_ trading compensation amounts. Figure 10a shows that the total CO_2_ trading compensation amounts increased significantly, from RMB 1055.17 million in 2003 to RMB 22,040.81 million in 2019. The area’s Gini coefficient first increased and then decreased, from 0.817 in the 2003–2006 period to 0.829 in the 2007–2013 period, and then fell to 0.789 in the 2014–2019 period, consistently above 0.5 (Figure 10c). This indicates a wide disparity in CO_2_ trading compensation amounts among municipalities. In particular, between 2007 and 2013, CO_2_ trading was highly dependent on a few economically advanced municipalities, such as Shanghai and Nanjing, increasing the vulnerability of the CO_2_ market. Since 2013, the vulnerability of the CO_2_ market decreased as the property rights and traceability of CO_2_ emissions became clear. Municipalities have become more active in CO_2_ trading and are generally more receptive to CO_2_ trading compensation amounts.

The CO_2_ trading data among municipalities showed that the CO_2_ surplus and deficit status were unstable between 2003 and 2006. This led to frequent role changes and complex game relationships among municipalities in the trading process, which was not conducive to government regulation. For example, Huaibei, Yancheng, and Suzhou (A.H.) gradually shifted from being purchasing municipalities to selling municipalities, whereas Bengbu, Yangzhou, and Chuzhou gradually shifted from being selling municipalities to purchasing municipalities. Between 2007 and 2013, more municipalities, such as Lianyungang and Wuhu, exceeded their CO_2_ allowances and shifted from selling municipalities to purchasing ones, whereas a few switched from purchasing municipalities to selling ones. During this period, the volatility of the YRD’s CO_2_ emissions efficiency led to high fluctuations in the CO_2_ unit prices. For example, the CO_2_ unit prices for selling municipalities such as Zhoushan declined due to a decline in CO_2_ emissions efficiency, which significantly reduced the profitability of CO_2_ trading in an environment where demand exceeded supply. The CO_2_-deficit municipalities such as Nanjing and Jiaxing had seen their CO_2_ emissions efficiency decline, leading to a gradual shift of the ‘right of first refusal’ to Ningbo and Wuxi, where CO_2_ emissions efficiency was rapidly increasing. Between 2013 and 2019, the trading roles, CO_2_ unit prices, tradable volumes, and rights of first refusal of municipalities all stabilized. In an environment in which demand exceeded supply for long periods, the game among CO_2_ market players was mainly played among purchasing municipalities. For example, Shanghai and Suzhou (J.S.) were in a race for the right of first refusal (lower CO_2_ buy-in unit price), whereas Bozhou and Fuyang were competing for purchasing rights to CO_2_ emission space.

## 5. Conclusions and Discussion

### 5.1. Conclusions

In this study, we estimated the CO_2_ footprints of 41 municipalities in the YRD based on nighttime lighting data and calculated their CO_2_ emissions efficiency using the super-SBM model. Based on this, we discussed the scenarios of horizontal CO_2_ compensation among municipalities from binary perspective—CO_2_ footprints and CO_2_ trading. Our main conclusions were as follows.

(1) With the relocation and upgrading of energy-intensive industries, the overall spatial correlation of CO_2_ footprints in the YRD has weakened. However, the HH and LL agglomeration patterns are still characterized by significant club convergence. Low-carbon economic development is a long-term process of a change in energy structure, with different municipalities located at various stages of evolution. Shanghai and others have experienced a decline in their CO_2_ footprints, whereas Hangzhou and others are in the transition phase toward a CO_2_ peak. The CO_2_ footprints of Taizhou (J.S.) and others are rising strongly, whereas Huangshan and others continue to maintain slow growth.

(2) The average value of CO_2_ emissions efficiency in the YRD has shown a fluctuating upward trend and a high degree of responsiveness to national policies. From 2000 to 2006, the southern Jiangsu and Hangzhou Bay regions were the growth hotspots of CO_2_ emissions efficiency, with Wuxi and Shaoxing showing relatively slow growth rates. From 2007 to 2013, the overall spatial pattern of CO_2_ emissions efficiency was high in the east and low in the west, with a symmetrical north–south distribution. From 2014 to 2019, the extreme high-value and high-value areas of CO_2_ emissions efficiency were distributed along the Shanghai–Wuhan and Zhuji–Yongjia highway axes, with the formation of a backbone structure of Shanghai–Nanjing and Shanghai–Hangzhou–Ningbo.

(3) In terms of CO_2_ footprint compensation amounts (baseline amounts + additional amounts), the ratio of CO_2_ footprint compensation to local GDP was less than 0.01% in most municipalities after 2011, and the trajectory of its standard elliptical center of gravity showed a regular shift. Therefore, it was appropriate for the YRD to implement a horizontal CO_2_ footprint compensation system after 2011. In the process, it would be necessary to strengthen ecological compensation for some municipalities in Anhui and implement dynamic adjustments for three years.

(4) In terms of CO_2_ trading compensation amounts, the CO_2_ emissions efficiency of most purchasing municipalities was higher than that of selling municipalities, which enhanced the economic added value of CO_2_ trading in the YRD. From 2000 to 2006, Huaibei and others exhibited fluctuating role transformations in CO_2_ trading, and the game relationship among municipalities was complex. From 2007 to 2013, the volatility of CO_2_ emissions efficiency led to high volatility in CO_2_ unit prices, reinforcing the fluidity of the ‘right of first refusal’ advantage. From 2014 to 2009, the role of municipalities, CO_2_ unit prices, tradable volume, and the right of first refusal gradually stabilized, the CO_2_ trading mechanism was improved, and the CO_2_ trading compensation amounts increased significantly.

### 5.2. Recommendations

Based on the above conclusions, we propose two potential routes for CO_2_ reduction: cross-border governance and the internal management of municipalities.

(1) We recommend the building of a closed-loop CO_2_ compensation management system consisting of ‘council + executive committee + regulator + industrial sector’ at the level of cross-border governance, rotating among three provinces and one provincial municipality. (i) The council should scientifically set the milestones of the ‘carbon peak’ and ‘carbon neutrality’ for each municipality and reasonably allocate CO_2_ allowances. At the same time, the council should establish an early-warning red line for the degradation of input/output indicators, as well as formulating policies for sharing responsibility and compensation for CO_2_ emission reductions. (ii) The executive committee should be responsible for financial transfer payments and horizontal compensation among local governments, according to whether the municipalities exceed their CO_2_ allowances. At the same time, it should regularly take stock of new trends in the development of low-carbon technologies and new changes in the CO_2_ emissions efficiency of municipalities and provide macro-guides for the inter-regional flow of production elements. (iii) The regulator should mainly supervise and arbitrate the decisions of the council and the ‘accountability and cash system’ of the executive committee, as well as managing the CO_2_ credits of each municipality. (iv) On the one hand, the industrial sector should regulate the industry and eliminate the problem of an underdeveloped production capacity. On the other hand, it should build a multilateral exchange platform to circumvent the problems of market information asymmetry and transaction hedging, as well as encouraging the organization of the main body to reach a green contract across the region.

(2) Each municipality should take into account its situation in order to develop a differentiated CO_2_ reduction route. In ‘high-efficiency and low-emissions’ municipalities, the labor- and energy-intensive industries showed outward decentralization. Compared with the industrial ‘carbon peak’, the reduction of CO_2_ emissions at the consumption end is relatively lagging. Therefore, these municipalities need to develop a broader carbon tax base. They should not only levy carbon taxes on energy-intensive industries, forcing industries to upgrade, but also gradually levy carbon taxes on residents to achieve a shift from a high-carbon society to a low-carbon society.

In ‘high-efficiency and high-emissions’ municipalities, the ‘CO_2_ decoupling’ phenomenon of economic development was not yet significant. Their higher CO_2_ emissions efficiency also relaxes the ecological constraints for these municipalities to take over the industrial transfer from ‘high-efficiency and low-emissions’ municipalities. However, if these municipalities only undertake purely replicative industries, this will introduce hidden problems for their sustainable development in later years. Therefore, these municipalities should levy a carbon tax on the production process, raise the entry threshold of energy-intensive industries, and improve the structure of domestic and international investment in the region.

For ‘low-efficiency and high-emissions’ municipalities, taking over resource-based industries is a vital choice to maximize resources and drive economic growth. However, resource-intensive industries are mostly high-carbon industries, which seriously hinder the development of a low-carbon economy. Therefore, these municipalities can decompose various by-products in the industrial chain level by level and form an efficient resource recycling system through the vertical and horizontal coupling of the industrial chain.

For ‘low-efficiency and low-emissions’ municipalities that are technologically underdeveloped and have a superior ecological background, a carbon tax should serve to build a firm ecological protection barrier. These municipalities should guide their industrial layouts and development in a centralized manner, and they should use their CO_2_ compensation amounts for the ecological transformation of industrial areas. They can also adopt the compensation method of ‘off-site development’ to find new development platforms in the neighboring ecological beneficiary municipalities, thus promoting the synergistic development of the local ecology and economy.

### 5.3. Discussion

Horizontal CO_2_ compensation studies have significant theoretical and practical implications for regional equity and sustainable development. The footprints method is an effective way to determine horizontal CO_2_ compensation. While accounting for CO_2_ footprints, Hu et al. introduced the ‘CO_2_ sink’ concept into the calculation, measured the difference between the two from the perspective of CO_2_ balance, and then multiplied it with the CO_2_ price to determine their CO_2_ compensation standards [9]. However, such compensation standards are based on the absolute amount of the CO_2_ deficit and do not consider the differences in resource endowments and emission reduction potential among different regions. The historical CO_2_ intensity method is a convenient solution to this problem. It calculates CO_2_ intensity based on municipalities’ historical production data and CO_2_ emissions, which are used as a basis to decrease year by year and thus determine the CO_2_ allowances for the following year. CO_2_ allowances can be adjusted in line with changes in input/output indicators, urging municipalities to reduce their own emissions [59]. Therefore, in this study, we applied the historical CO_2_ intensity method to allocate CO_2_ allowances to the 41 municipalities, as well as differentially levied CO_2_ compensation amounts based on whether the CO_2_ footprints exceeded their allowances. This approach differs from those presented in the studies of Hu et al. [9] and Yang et al. [10] in that they only levied amounts on CO_2_/environmental-deficit cities, probably due to the consideration of reducing the economic burdens on CO_2_/environmental-surplus cities. In this study, the CO_2_ emission rights within the CO_2_ allowances maintain sustainable economic development, and the government levies the CO_2_ footprint baseline compensation amounts to each municipality at a lower baseline CFIE (*δ_q_*). In addition, the CO_2_-deficit municipalities need to pay additional CO_2_ footprint amounts at a higher additional CFIE (*δ*_∆*q*_) to compensate CO_2_-surplus municipalities, which reduces the distortionary effect of horizontal compensation to a certain extent. Moreover, in this study we also considered the ratio of CO_2_ footprint compensation to local GDP and the stability of the CO_2_ surplus and deficit status. When the ratio is too high or when the surplus and deficit status is unstable, horizontal CO_2_ compensation is not recommended.

In addition, we considered the added value and fairness of CO_2_ trading with a price mechanism determined by CO_2_ emissions efficiency. Shi et al. focused on the operational mechanisms of inter-city CO_2_ trading, the allocation model of CO_2_ allowances, and the impact of market supply and demand on CO_2_ emissions [59]. However, their study did not discuss the economic value added to CO_2_ emission rights in CO_2_ trading. In response, our study represents a continuation of this research. As municipalities have different CO_2_ emission efficiencies, the economic benefits generated per unit of CO_2_ emissions vary. CO_2_-deficit municipalities require the purchase of CO_2_ emissions rights from CO_2_-surplus ones. CO_2_ emission rights for trading receive a higher economic value if the CO_2_ emissions efficiency of the purchasing municipalities is higher than that of the selling ones. Although the selling municipalities receive turnover for the sale of CO_2_ emissions rights, the purchasing ones acquire a higher added value. Therefore, the purchasing municipalities are required to pay CO_2_ trading compensation amounts. This approach helps to establish a mechanism to share the responsibility for emissions reductions among the municipalities in the YRD and provides a model for a horizontal CO_2_ compensation system for the whole country.

Ultimately, we believe it is feasible to establish a mechanism for apportioning responsibility for emissions reductions from both a ‘CO_2_ footprint’ and ‘CO_2_ trading’ perspective. There is strong empirical evidence that some countries are implementing a combination of the CO_2_ tax and CO_2_ trading. For example, Denmark, Finland, and Norway have introduced CO_2_ taxes and followed them up with the EU Emissions Trading Scheme. Although such policies have targeted enterprises or industries, they have inspired China to develop regional policies to reduce emissions. For example, the Opinions on Improving the Compensation Mechanism for Ecological Protection mentioned establishing a system for the initial allocation of CO_2_ emissions rights and promoting horizontal ecological protection compensation. The Study on China’s CO_2_ Balance Trading Framework recommended introducing an inter-provincial CO_2_ emissions trading system. This paper provides a methodological basis for developing innovative regional CO_2_ reduction policies.

There are still shortcomings of this study. The optimal CO_2_ footprint compensation amounts depended on the marginal damage or marginal pollution cost per unit of pollutant, determined at the optimal pollution emission level. However, in reality, this cost is difficult to measure, and to some extent, it affects the setting of the optimal CFIE (*δ_q_* and *δ_∆q_*). In this study, we initially established the CFIE with a 3-year adjustment cycle and the value-added coefficient for CO_2_ turnover based on the VAT rate for intangible asset transfers. Refinements and development should be carried out in subsequent studies in order to reach a more desirable CO_2_ reduction state. There is also a need to further clarify the municipalities’ CO_2_ emission reduction targets and set entry thresholds for CO_2_ trading.

## Figures and Tables

**Figure 1 ijerph-20-01369-f001:**
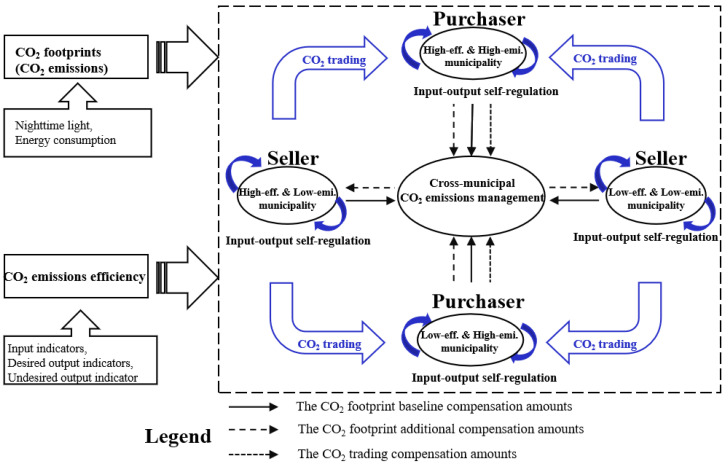
Diagram of the horizontal CO_2_ compensation mechanism in the YRD. Note: eff. and emi. are abbreviations for efficiency and emissions, respectively.

**Figure 2 ijerph-20-01369-f002:**
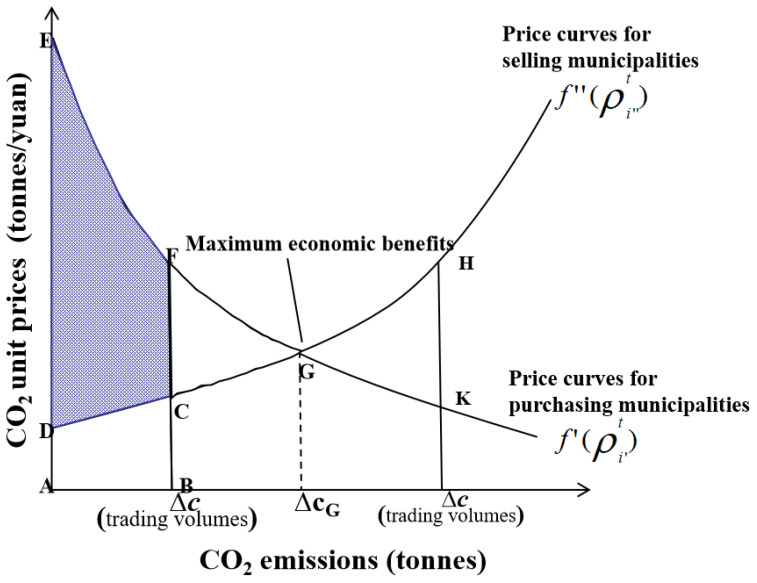
Diagram of the economic benefits based on CO_2_ trading. Note: When ∆*c* < ∆*c_G_*, the CO_2_ emissions efficiency of all the purchasing municipalities is higher than that of the selling ones, and CO_2_ emission rights for ∆*c* tons lead to positive economic benefits in CO_2_ trading (*S_DCEF_*), which is also the overall economic benefit. when ∆*c* > ∆*c_G_*, the CO_2_ emissions efficiency of some purchasing municipalities is lower than that of the selling ones, and ∆*c* tons of CO_2_ emission rights generate positive benefits (*S_EDG_*), while (∆*c* − ∆*cG*) tons lead to negative economic benefits (*S_HKG_*), and the overall economic benefit is (*S_EDG_* − *S_HKG_*).

**Figure 3 ijerph-20-01369-f003:**
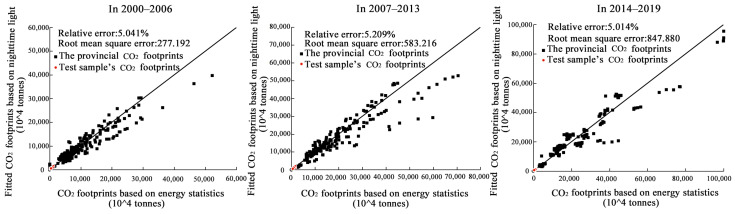
Deviation of the fitted CO_2_ footprints based on night-time light from the CO_2_ footprints based on energy statistics.

**Figure 4 ijerph-20-01369-f004:**
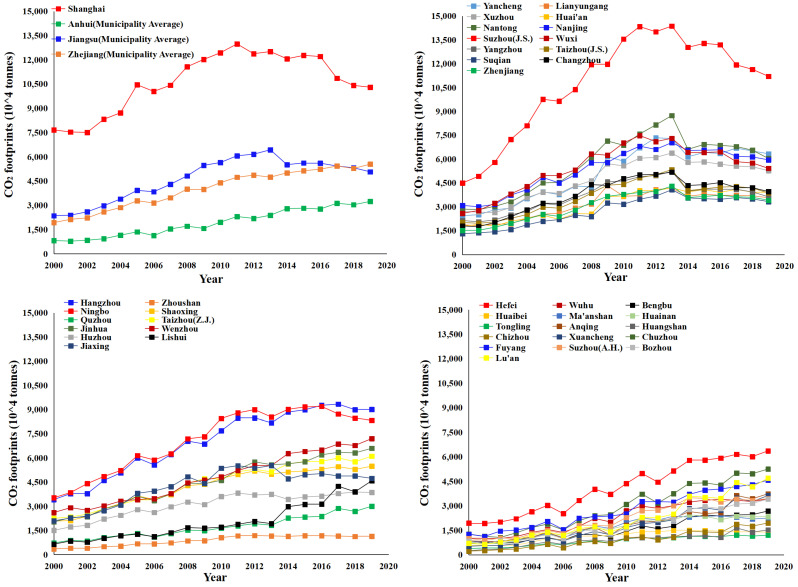
Spatio-temporal evolution of the CO_2_ footprints in the YRD from 2000 to 2019.

**Figure 5 ijerph-20-01369-f005:**
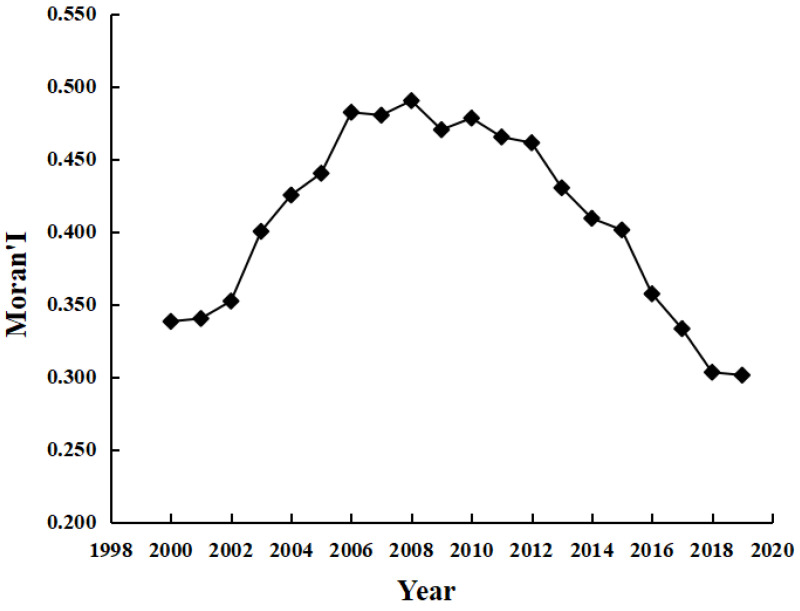
Global spatial autocorrelation Moran’s I of CO_2_ footprints in the YRD.

**Figure 6 ijerph-20-01369-f006:**
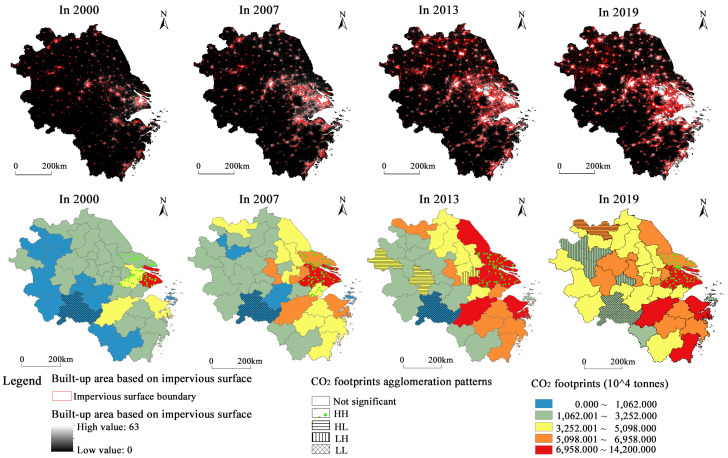
Spatial distribution of night light and CO_2_ footprints in the built-up area of the YRD.

**Figure 7 ijerph-20-01369-f007:**
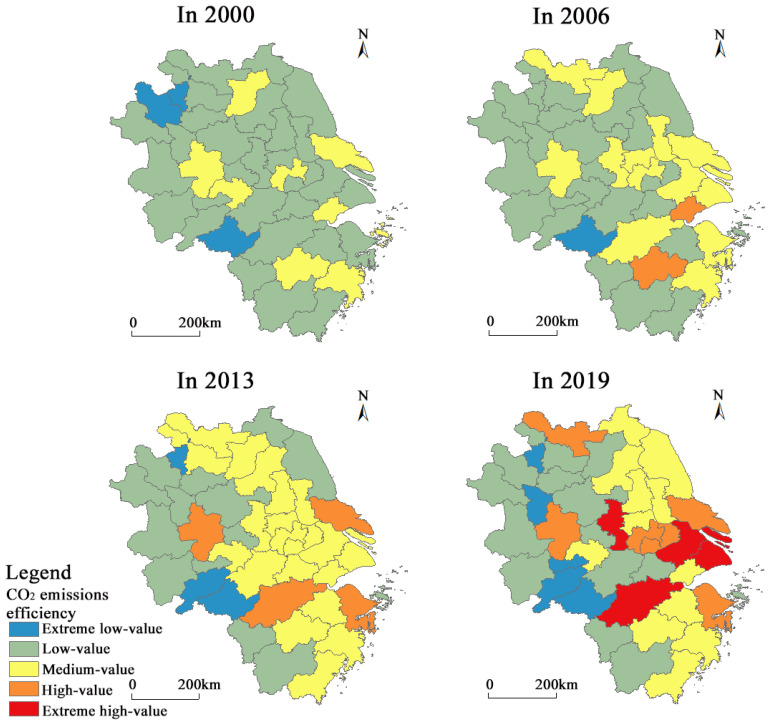
Spatio-temporal evolution of CO_2_ emissions efficiency in the YRD.

**Figure 8 ijerph-20-01369-f008:**
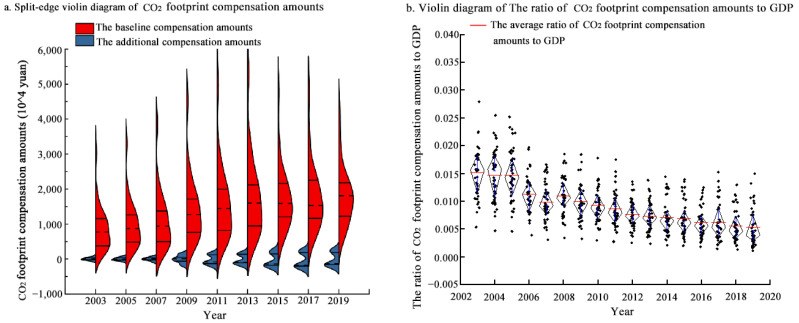
CO_2_ footprint compensation amounts and its ratio to local GDP in the YRD.

**Figure 9 ijerph-20-01369-f009:**
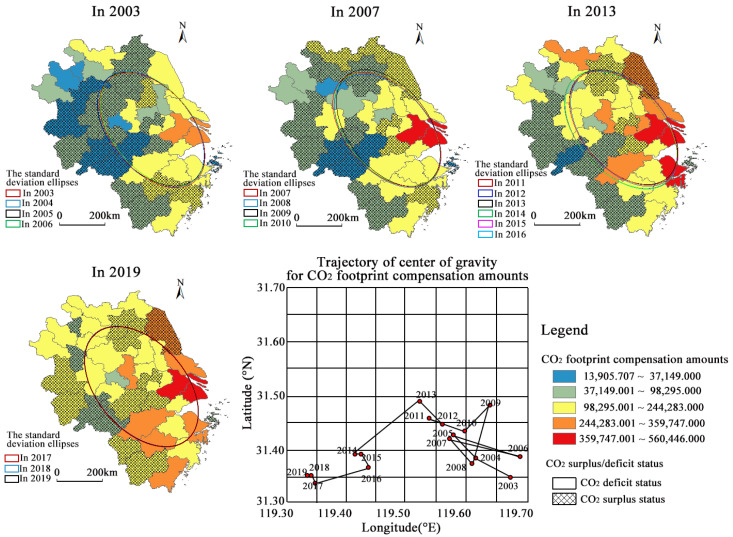
Evolution of CO_2_ surplus and deficit and CO_2_ footprint compensation amounts in the YRD.

**Figure 10 ijerph-20-01369-f010:**
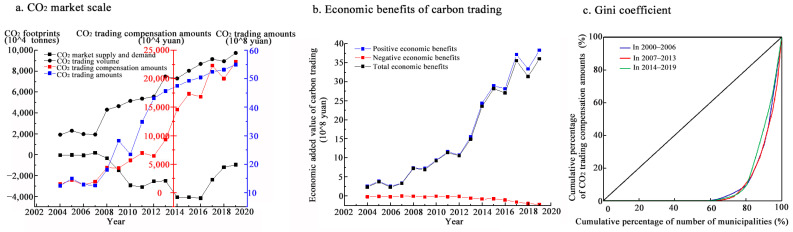
Evolution of supply–demand relationship and economic benefits in CO_2_ trading of the YRD.

**Table 1 ijerph-20-01369-t001:** Parameters and estimation accuracy of the CO_2_ footprints fitting curve.

Periods	*ω*	*τ*	*R* ^2^	Relative Error
2000~2006	0.035	1	0.801	5.041%
2007~2013	0.036	1	0.799	5.209%
2014~2019	0.032	1	0.807	5.014%

Note: In determining the CO_2_ footprints fitting curve, *τ* was set to 1/2, 1, 2, and 3 separately. The results showed that the *R*^2^ of the fitting curve was optimal when *τ* = 1 (linear fit). Therefore, only the results for *τ* = 1 are listed in the table.

## Data Availability

No new data were created or analyzed in this study. Data sharing is not applicable to this article.

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
