# Peer review of "Horizontal CO2 Compensation in the Yangtze River Delta Based on CO2 Footprints and CO2 Emissions Efficiency"

_ijerph, 2023, doi:10.3390/ijerph20021369_

Round 1

Reviewer 1 Report

This study employed nighttime light data to estimate the municipal CO2 footprints in the YRD from 2000 to 2019, measured CO2 emissions efficiency using a super SBM model, and designed a scenario of inter-city CO2 compensation. This paper is interesting but still has some problems as follows: 

#1 This paper is in poor language and should be copyedited by a native speaker. 

#2 Figures are in low quality and hard to capture the necessary information. 

#3 The font size in equations seems a little outfit, and should be modified. 

#4 Literature review should be further perfected, for example, regional carbon footprints are generally estimated by MRIO models. In addition, a comparison between CO2 footprints derived from different methods should be concluded to illustrate their strengths, weakness and applicability. 

#5 Some relevant studies from MDPI publisher should be citied, for example, "Analysis of CO2 emission performance and abatement potential for municipal industrial sectors in Jiangsu, China." Sustainability 8.7 (2016): 697. 

#6 Discussion should be separated from Conclusion and necessary policy implications should be provided based on the findings.

Reviewer 2 Report

In order to clarify the CO2 emission reduction in the Yangtze River Delta region and design a mechanism for sharing responsibility across municipalities, this paper predicts the CO2 emission rate using the SBM model based on nighttime lighting data to estimate the CO2 footprint of the Yangtze River Delta region from 2000 to 2019. This paper builds a well-fitting model based on reality, and the prediction results are accurate and helpful conclusions are obtained. I have only one main question and some minor questions for the authors, and I am willing to recommend this article for publication if the authors can make reasonable responses and corrections.  

A main question I have is that I am concerned that the data for this paper's study is from 2000-2019, is the data not quite timely for a paper that will be published in 2022, or what is the basis for the authors' selection of data in this way?  

There are some minor questions below that require further responses from the author  

1. The abstract section needs to further clarify the innovation and research significance of this study, and the authors are invited to supplement the abstract section.  

2. I am not sure if it is a problem with the review file or with the figures. Anyway, the clarity of Figure 1 is low, and since it contains a lot of textual content, the authors are expected to further check the resolution of the images for clarity. Similar issues exist with other figures, and the authors are expected to further check the clarity of the figures in the formal manuscript.

 3. In figure 3, the authors chose the data for the period 2000-2019 to determine the fitted curves to estimate the CO2 footprint of the city. In any case, it is worth noting that the data clearly maintain three different trends and growth rates for the period 2007-2013, while the authors only considered a single linear fit.  

4.The inverted U-shaped trend of the data in Table. 2 is worthy of attention, but the inverted U-shape shown only in the form of a table may not be intuitive enough, and the form of a graph may be better to show the trend.

Reviewer 3 Report

Manuscript ID: ijerph-2076486

Title: Horizontal CO2 Compensation in the Yangtze River Delta Based on CO2 Footprints and CO2 Emissions Efficiency

This article estimated the municipal CO2 footprints (CO2 emissions) in the YRD from 2000 to 2019 based on night-time light data and measured CO2 emissions efficiency using a super SBM model. Based on this, this article designed a scenario of horizontal CO2 compensation among the YRD’s municipalities from the perspectives of both CO2 footprints and CO2 trading (CO2 unit prices in trading are determined by CO2 emissions efficiency).

 I appreciate the opportunity to review the proposed article and hope that my considerations will help to improve the work. My detailed comments are as follows:

Abstract: The abstract section should clearly mention the objective, methods, and concise results with sequence. Revise accordingly

Introduction: In the introduction section, research objectives are missing. Clarify the background, objectives, research gaps, and innovations in this section.

Literature review and study framework: Add more appropriate and advanced literature on the environmental efficiency. Add some literature on environmental efficiency measured through DEA, you could get help from the following studies to add more recent literature on EE.

 https://doi.org/10.1002/csr.2014

https://doi.org/10.1016/j.fuel.2021.123094

https://doi.org/10.1016/j.fuel.2021.123098

https://www.sciencedirect.com/science/article/pii/S2352484722014263

Data and methods: The super-SBM model based on undesired outputs is employed to measure the efficiency. Although the technique is advanced and accurate, add more explanation to justify the application of this model. Further, selecting inputs and outputs is vital in DEA efficiency estimation. Mention the studies which you used to select the variables.

Conclusion; Add limitations and future research ideas in the conclusion section. Expand the policy implications for China.

This version of the manuscript contains many grammatical and typo errors; revise the manuscripts for these concerns.

Round 2

Reviewer 1 Report

The authors have answered my comments.